# NOTCH Activation via gp130/STAT3 Signaling Confers Resistance to Chemoradiotherapy

**DOI:** 10.3390/cancers13030455

**Published:** 2021-01-26

**Authors:** Kristin Koerdel, Melanie Spitzner, Thomas Meyer, Niklas Engels, Florian Krause, Jochen Gaedcke, Lena-Christin Conradi, Martin Haubrock, Tim Beißbarth, Andreas Leha, Steven A. Johnsen, B. Michael Ghadimi, Stefan Rose-John, Marian Grade, Jürgen Wienands

**Affiliations:** 1Institute of Cellular and Molecular Immunology, University Medical Center Goettingen, 37073 Goettingen, Germany; kristin.koerdel@med.uni-goettingen.de (K.K.); nengels@gwdg.de (N.E.); 2Department of General, Visceral and Pediatric Surgery, University Medical Center Goettingen, 37075 Goettingen, Germany; melanie.spitzner@med.uni-goettingen.de (M.S.); florian.krause@stud.uni-goettingen.de (F.K.); jochen.gaedcke@med.uni-goettingen.de (J.G.); lena.conradi@med.uni-goettingen.de (L.-C.C.); johnsen.steven@mayo.edu (S.A.J.); mghadim@uni-goettingen.de (B.M.G.); 3German Centre for Cardiovascular Research, Department of Psychosomatic Medicine and Psychotherapy, University Medical Center Goettingen, 37073 Goettingen, Germany; thomas.meyer@med.uni-goettingen.de; 4Institute of Medical Bioinformatics, University Medical Center Goettingen, 37073 Goettingen, Germany; martin.haubrock@bioinf.med.uni-goettingen.de (M.H.); tim.beissbarth@bioinf.med.uni-goettingen.de (T.B.); 5Department of Medical Statistics, University Medical Center Goettingen, 37073 Goettingen, Germany; andreas.leha@med.uni-goettingen.de; 6Institute of Biochemistry, Christian-Albrechts-University, 24098 Kiel, Germany; rosejohn@biochem.uni-kiel.de

**Keywords:** treatment resistance, chemoradiotherapy, STAT3, NOTCH, gastrointestinal cancer

## Abstract

**Simple Summary:**

Resistance to chemoradiotherapy represents a fundamental problem in modern oncology because it exposes patients to the potential negative side-effects of both radiation and chemotherapy without any clinical benefit. This study uncovers that the inflammatory signaling hub STAT3 conspires with the cell fate regulator NOTCH in rendering tumor cells refractory to chemoradiotherapy. The dichotomic signal alliance is based on a so-far unknown STAT3 target gene, RBPJ, providing the transcriptionally active partner of NOTCH intracellular domain. Unexpectedly, the latter is permanently produced by tonic proteolysis. Tumor mouse models and cancer patient cohorts demonstrate the usefulness of the STAT3/NOTCH axis as biomarker for patient stratification, and importantly, that STAT3 inhibition is a promising treatment option for re-sensitization of CRT-refractory tumors.

**Abstract:**

Resistance of tumor cells to chemoradiotherapy represents a fundamental problem in clinical oncology. The underlying mechanisms are actively debated. Here we show that blocking inflammatory cytokine receptor signaling via STAT3 re-sensitized treatment-refractory cancer cells and abolished tumor growth in a xenograft mouse model when applied together with chemoradiotherapy. STAT3 executed treatment resistance by triggering the expression of RBPJ, the key transcriptional regulator of the NOTCH pathway. The mandatory RBPJ interaction partner, NOTCH intracellular domain, was provided by tumor cell-intrinsic expression of NOTCH ligands that caused tonic NOTCH proteolysis. In fact, NOTCH inhibition phenocopied the effect of blocking STAT3 signaling. Moreover, genetic profiling of rectal cancer patients revealed the importance of the STAT3/NOTCH axis as NOTCH expression correlated with clinical outcome. Our data uncovered an unprecedented signal alliance between inflammation and cellular development that orchestrated resistance to chemoradiotherapy. Clinically, our findings allow for biomarker-driven patient stratification and offer novel treatment options.

## 1. Introduction

At present, chemoradiotherapy (CRT) plays an integral part in treatment concepts for various tumor entities. Regarding cancers of the rectum, locally advanced stages of this disease are treated with preoperative CRT followed by radical surgical resection [1,2,3]. However, about one third of patients will have no or little response to preoperative CRT [1,3,4]. Hence, patients with resistant tumors show no benefit from the treatment but are afflicted with the potential acute and long-term side effects of both chemotherapy and radiation, which include hematologic, gastrointestinal, genitourinary, and dermatologic toxicity [1,3,4,5,6]. Importantly, poor response to preoperative CRT directly correlates with an impaired overall survival [7,8]. Therefore, re-sensitization of tumor cells that are partially or even fully refractory to CRT represents an attractive solution to a fundamental clinical and socioeconomic problem in oncology.

However, the molecular basis of CRT resistance is complex and polymodal. While evidence suggests that the underlying mechanisms include, for instance, altered cell cycle regulation, immune evasion, hypoxia, evasion of apoptosis, or the existence of resistant tumor subclones, effective therapeutic strategies have not yet been incorporated into the clinical setting [9,10,11,12,13,14]. Our group previously demonstrated a potential role of Signal Transducer and Activator of Transcription 3 (STAT3) in mediating CRT resistance of colorectal cancer (CRC) cell lines. We now concentrated on the molecular endowments of STAT3-controlled treatment resistance and uncovered that CRT-resistant tumor cells are equipped with autonomous NOTCH signaling activity. An active heterodimeric NOTCH effector complex, called RBPJ/NICD, was assembled in CRT-resistant tumor cells as a result of tonic signal input from two separate signaling cascades. The transcriptionally active component Recombination Signal Binding Protein for Immunoglobulin k J-region (RBPJ) was induced by inflammatory cytokine receptor signaling and phosphorylation of STAT3. The second component, NOTCH intracellular domain (NICD), resulted from tonic cleavage of NOTCH surface protein caused by the endogenous and tumor cell-intrinsic expression of NOTCH ligands together with active NOTCH-processing proteases. In essence, a hitherto unknown alliance between inflammation and developmental pathways converges at the level of RBPJ and NICD and blocks CRT responsiveness. We furthermore show that preventing RBPJ/NICD complex formation re-sensitized CRT-refractory tumor cells and thus offers a promising therapeutic strategy to solve the problems that come along with CRT resistance in rectal cancer.

## 2. Results

### 2.1. Inflammation Promotes CRT Resistance

STAT3 is the intracellular signal gate keeper of a variety of cell surface receptors with inflammatory activity and is involved in numerous disease entities, including colorectal cancer [15,16,17,18]. Our previous observation that diminished expression of STAT3 in CRC correlated with increased CRT sensitivity indicated that inflammatory signaling may account for CRT resistance even though at the time neither upstream inducers nor downstream effectors of STAT3 were functionally assigned to CRT resistance [19]. Appendix A shows that not only mere expression of STAT3 but also its robust phosphorylation and high transcriptional activity were associated with increased survival of CRC cells treated with CRT. To directly assess whether these observations are causes or consequences of CRT resistance, we tested whether the gain of STAT3 activity converts CRT-sensitive into CRT-resistant cells. LS411N cells that are STAT3-deficient and CRT-sensitive (Appendix A, left panel) were reconstituted with either wild-type STAT3, or signaling-inactive versions, in which critical tyrosine and/or serine phosphorylation sites were inactivated by replacement with phenylalanine or alanine, respectively. Expression of wild-type, but not mutant STAT3, restored STAT3 transcriptional activity in LS411N cells (Figure 1A, upper left and right panel). Importantly, the presence of wild-type STAT3 increased survival in a colony formation assay (CFA) (Figure 1A, lower panel), while expression of signaling inactive mutants did not (Appendix A). These data revealed a direct contribution of STAT3 to CRT resistance and strongly indicated an input of upstream regulatory signals.

Potent activators of STAT3 are inflammatory cytokine receptors such as the receptor for IL-6, which utilizes the common gp130 signaling component and its associated Janus tyrosine kinases (JAK) to phosphorylate and thereby activate STAT3 [15,16,17,18,20,21] (see also Appendix A). To assess the influence of the gp130/JAK signaling cascade on CRT resistance, we treated the human rectal cancer cell lines SW837 and SW1463, both of which are STAT3-positive and CRT-resistant with a chimeric fusion protein, called Hyper-IL-6, which encompasses IL-6 and the soluble IL-6 receptor chain and therefore mimics IL-6 signaling via membrane-bound and soluble IL-6 receptor [22] (see also Appendix A). Stimulation of both SW837 and SW1463 cells with Hyper-IL-6 triggered robust STAT3 phosphorylation on tyrosine 705, which translated into increased STAT3 transcriptional activity, along with prolonged CFA survival after treatment with CRT (Figure 1B). To further investigate the role of inflammatory signals on CRT resistance, we employed established inhibitors of the IL-6 receptor signaling cascade. Tocilizumab is a clinically used monoclonal antibody that binds to the IL-6 receptor and thereby inhibits its ligation-induced signal output [16,17,21]. Treatment of SW837 and SW1463 cells with tocilizumab dampened STAT3 tyrosine phosphorylation as well as its transcriptional activity, and, consequently, rendered both cell lines more sensitive to CRT, as revealed by their decreased CFA survival rates (Figure 1C, left and right panel). Likewise, the use of ruxolitinib, a potent JAK inhibitor [16,17,21], reduced STAT3 signaling and sensitized SW837 and SW1463 cells to CRT (Figure 1D, left and right panel). Note that none of the described treatments impacted on CFA survival of STAT3-deficient LS411N cells, which served as negative control in all assays (Appendix A). In summary, the extent of CRT resistance in human rectal cancer cells could be tuned in both directions by manipulating the activity of the IL6/STAT3 signaling pathway.

### 2.2. Inhibition of the gp130/STAT3 Axis as a Therapeutic Strategy

To evaluate a potential clinical applicability of our findings, we tested whether inhibition of the gp130/STAT3 signal axis with already established compounds impacts on CRT resistance and whether it can suppress the growth of tumor transplants under CRT in a living organism. To this end, we chose napabucasin (BBI608), a small-molecule inhibitor of STAT3 [23], which has already been tested in a phase-III clinical trial for highly advanced, chemotherapy-refractory CRC [24], and which can be administered orally. Indeed, napabucasin treatment of SW837 and SW1463 cells suppressed tyrosine phosphorylation and transcriptional activity of STAT3 and concomitantly sensitized both cell lines to CRT in our CFA survival assay without affecting the amount of STAT3 expression (Figure 2A, left and middle panels). In accordance with STAT3 deficiency, the CFA survival of LS411N cells remained unaffected by napabucasin (Appendix A). To assess potential off-site effects of napabucasin action, we combined that treatment with RNAi against STAT3 in SW1463 cells. As observed before, both approaches individually impinged on CRT sensitivity and their combination did not have synergistic effects (Figure 2A, right panel). Thus, the effect of napabucasin in our cell culture assays can specifically be ascribed to inhibition of STAT3 tyrosine phosphorylation, even though the exact mode-of-action of napabucasin has not yet been fully explored [23].

To corroborate that finding in vivo, we used a xenograft model, in which human SW1463 rectal cancer cells were transplanted into immunodeficient *Foxn1^-/-^/*nude mice, followed by a treatment protocol that closely recapitulates clinical conditions, i.e., fractionated doses of both radiation and chemotherapy (Figure 2B). We firstly demonstrated that the body weight of mice remained stable, regardless of the treatment modalities, indicating a low overall toxicity (Figure 2C). Treatment with napabucasin alone did not suppress the growth of tumor transplants compared to treatment with DMSO alone (Figure 2D). However, when combined with CRT, napabucasin treatment completely abrogated tumor growth during treatment (Figure 2E). The effect of napabucasin on tumor growth was associated with almost absent phosphorylation of STAT3 in the tumor cells despite normal STAT3 expression (Appendix A).

An important clinical aspect is the regrowth of tumors after the end of medical treatment. To address that issue, we monitored the tumor volume in our mice for an additional period of 23 days, i.e., until day 45 after treatment start, which represents the time point when the first mouse died in the DMSO-treated control group. Figure 2F shows that in napabucasin-treated mice, tumor regrowth was significantly inhibited. In addition, treatment with napabucasin significantly increased the time to tumor tripling (Figure 2G), which represents an established criterion to assess full tumor regrowth [25].

We next tested whether targeting the gp130/STAT3 signaling axis also represents a potential clinical strategy for other tumor entities. To this end, we focused on esophageal cancer because preoperative CRT represents a common treatment regime for patients with locally advanced stages of this disease [26]. Most esophageal cancer cell lines showed prominent expression of STAT3 and some of them even showed constitutive phosphorylation of STAT3 (Figure 3A). Treatment with napabucasin resulted in a significant sensitization to CRT, both in the adenocarcinoma cell line FLO-1 (Figure 3B) and in the squamous cell carcinoma cell line Kyse-150 (Figure 3C). Of note, the effect was more pronounced in the squamous cell carcinoma cell line, which may be due to more effective inhibition of STAT3 phosphorylation (Figure 3C, left). Together, these data suggest that inhibition of the gp130/STAT3 signaling axis may be applied as therapeutic measure in several entities of CRT-resistant cancers. Furthermore, phosphorylation of STAT3 may be used as a parameter for patient stratification and subsequent treatment (Figure 3D).

### 2.3. Resistance-Associated STAT3 Target Genes

To now delineate how inflammatory signals control CRT resistance, we analyzed how STAT3 pathway perturbation affects the global transcriptional activity of rectal cancer cells. Using next-generation RNA sequencing (RNA-Seq; Figure 4A), we determined differential gene expression profiles of SW837 cells under three experimental conditions: (a) cellular stimulation with Hyper-IL-6 either in the presence of STAT3, or (b) upon siRNA-mediated STAT3 silencing, and (c) targeted STAT3 expression without further stimulation (Figure 4B,C). Setting (a) is aimed at the identification of genes whose expression is per se affected by cytokine receptor signaling, while settings (b) and (c) addressed the importance of STAT3 for the up- or down-regulation of genes under stimulatory or inhibitory conditions, respectively. Phosphorylation of STAT3 was tested for all conditions by immunoblotting (Appendix A). The analysis of the three individual settings revealed 231 genes (siCtrl. vs. siCtrl. + Hy-IL-6), 2969 genes (siCtrl. vs. siSTAT3), and 3738 genes (siCtrl. + Hy-IL-6 vs. siSTAT3 + Hy-IL-6), respectively, that were differentially regulated with a false discovery rate (FDR) < 0.05 (Figure 4B,C). Overrepresented, biologically annotated pathways are summarized in Appendix A.

A total number of 71 genes was significantly up- or down-regulated in all three settings (Figure 4B, yellow). This indicates that the altered transcriptional activity of these 71 genes is dually influenced by STAT3 expression and cellular stimulation. In order to technically validate these results, the expression levels of 12 selected genes were quantified for all three conditions using qPCR analysis. Data obtained by RNA-Seq tightly correlated with those generated by qPCR, demonstrating the reliability of our screening approach (Figure 4D). Next, we stringently filtered for genes that were upregulated after pathway stimulation with Hyper-IL-6, and simultaneously but inversely, downregulated after STAT3 inhibition, and vice versa. This kind of Opposite Direction Analysis (ODA) resulted in 55 candidate genes that are likely to play a prominent role for STAT3-mediated CRT resistance (Figure 4E and Appendix A). Besides *STAT3* itself, our ODA revealed *SOCS3*, a negative feedback regulator of JAK-STAT signaling [27], and *ELF3*, a transcription factor associated with Wnt/β-catenin signaling, which represents a key oncogenic pathway previously linked to CRT resistance [28,29]. Other putative STAT3 targets were *DPYD*, a gene encoding a key 5-FU-metabolizing enzyme [30], *MUC1*, which impacts on the response to radiotherapy in pancreatic cancer [31], and *HIF1A*, an established target of JAK-STAT signaling and previously reported as potential determinant of tumor radiosensitivity [32]. Unexpectedly, our analysis also uncovered *RBPJ*, the key transcriptional regulator of the NOTCH pathway. Following ligation of NOTCH on the cell surface by DELTA/Jagged ligands, NICD becomes proteolytically cleaved and assembles with RBPJ in the nucleus to drive expression of NOTCH target genes [33,34] (for pathway details, see Appendix A).

### 2.4. Convergent STAT3 and NOTCH Signaling Cause CRT Resistance

To directly assess a functional relationship between inflammatory signaling and cell developmental (NOTCH) pathways, we first analyzed whether *RBPJ* is a direct target of STAT3 by Electrophoretic Mobility Shift Assay (EMSA). In fact, we identified a canonical docking site for STAT family members, called interferon-gamma activated sequence (GAS) [35], in the first intron of the human *RBPJ* locus approximately 300 bp 3′ of the known promotor region (Appendix A). EMSA revealed robust binding of radioactively labeled GAS probes from the *RBPJ* promotor to STAT proteins of Hyper-IL-6-stimulated SW837 cells, whereas the mutated control GAS probe showed no binding (Figure 5A). This identified *RBPJ* as a direct STAT3-regulated target gene.

Importantly, when RBPJ expression was silenced in otherwise CRT-resistant SW837 rectal cancer cells, the cells were re-sensitized to CRT (Figure 5B), showing that RBPJ—like STAT3—is a key determinant of CRT resistance. Moreover, RBPJ silencing phenocopied STAT3 silencing as silencing of RBPJ alone was as effective as inhibition of STAT3, and the combined interference with both proteins had no additive effect on CRT re-sensitization. In fact, the CFA survival curves of all three experimental conditions were virtually identical (Figure 5B). Hence, RBPJ is not just one of many but rather is a direct and major effector of STAT3-controlled CRT resistance. This conclusion is further supported by the observation that sensitization to radiotherapy (RT) following RNAi-mediated inhibition of RBPJ was even more pronounced after prior stimulation with Hyper-IL-6 (Appendix A).

RBPJ is best known for being the mandatory binding partner of NICD to constitute the transcriptionally active effector complex of NOTCH-activated cells [33,34]. In the absence of NICD, however, RBPJ has been reported to act as transcriptional repressor [33,37]. It was thus important to test for the presence of NICD in cells that are resistant or sensitive to CRT. Anti-NICD immunoblot analysis revealed a robust NICD signal in CRT-resistant SW837 and SW1463 cells (Figure 5C, upper panel). By contrast, the CRT-sensitive line LS411N did not express NICD (Figure 5C, upper panel). Furthermore, in accordance with the presence or absence of NICD, expression of the transcription factor HES1, a main target of active NOTCH signaling [38], was weak in LS411N cells but readily detected in SW837 and SW1463 cells with signal intensities that are proportional to the NICD positivity and CRT sensitivity of these cells (Figure 5C, upper panel). Finally, the presence of NICD in the tested cell lines directly correlated with the transactivation activity of STAT3 (Figure 5C, lower panel). Thus, as a consequence of inflammatory STAT3 signaling NICD and RBPJ form a functional transcription factor complex in CRT-resistant rectal cancer cells. Importantly, the amounts of NICD and RBPJ in CRT-resistant SW837 and SW1463 cells increased upon irradiation, whereas they decreased in CRT-sensitive cell line LS411N (Figure 5D). This observation indicates that irradiation of already CRT-resistant rectal cancer cells even further promotes their resistance by inducing the generation of NICD and RBPJ. Thus, CRT treatment of patients who have a CRT-resistant form of rectal cancer may be a contra-productive and even harmful measure.

Next, we investigated the cause of constitutive NICD production in CRT-resistant cells by a comprehensive expression analysis of proteins that regulate NOTCH processing (Figure 5E). Our cells were tested positive for different patterns of NOTCH ligands (Jagged 1/2 and DELTA-like) and NOTCH cleaving components such as ADAM proteases and presenilins (γ-secretases). However, a combination of proteins capable of processing NOTCH was found only in CRT-resistant SW837 and SW1463 cells, but not in CRT-sensitive LS411N cells. These observations indicated a cell-intrinsic tonic NOTCH processing activity that might be relevant for CRT resistance. To test this hypothesis, we inhibited the activity of γ-secretases using the chemical compound DAPT. Indeed, DAPT treatment re-sensitized SW837 cells to CRT (Figure 5F). Moreover, treatment with DAPT resulted in a sensitization to CRT similar to that seen when RBPJ was silenced by RNAi, while the combined blockade of the γ-secretases complex and RBPJ had no additive effect (Figure 5F).

Finally, to test if NOTCH family members indeed represent relevant molecular elements in human rectal cancer, we analyzed pretherapeutic gene expression profiles obtained from 207 patients with locally advanced rectal cancer who were treated with preoperative CRT. In fact, high expression of NOTCH2, NOTCH3, and NOTCH4 in the tumors of our patient cohort was associated with impaired disease-free survival, while NOTCH1 had no effect (Figure 5G and Appendix A). These data illustrate the strong correlation between NOTCH family proteins in rectal cancers and CRT resistance in a clinical context. In combination with our detailed molecular analyses, they strongly indicate that the interplay between inflammatory gp130/STAT3 signaling and the NOTCH pathway has a central role in mediating CRT resistance in rectal cancer.

## 3. Discussion

The mechanisms that underlie CRT-resistance, both tumor-intrinsic as well as tumor-extrinsic ones, are actively debated [9,10,11] including the functional role of STAT3 [19]. We now show that the tonic activity of STAT3 in rectal cancer cells is key to their resistance to CRT. STAT3 facilitates CRT resistance by inducing the expression of RBPJ, which represents the central effector protein of the NOTCH pathway [33,34]. This cell-autonomous signal axis could be further potentiated by triggering cytokine receptors of the gp130 family, an event that may also happen in the context of an inflammatory microenvironment found in many solid tumors [10,39]. Vice versa, inhibition of the gp130/STAT3 signaling at different levels re-sensitized otherwise resistant rectal cancer cells to CRT in our cell culture systems as well as in our in vivo xenograft tumor model. This mouse model has the advantages that it not only mirrors the clinical setting of fractionated doses of both irradiation and chemotherapy, but also allows for focusing on tumor-intrinsic factors. Noteworthy, the role of tumor-extrinsic factors and the impact of the tumor micromileu on CRT resistance can now be studied by additional interesting model systems that have recently been reported by other labs [40,41].

Our results confirm and extend the recent observation from Nagaraju and colleagues, who demonstrated a sensitization to CRT of HCT116 colon cancer cells following treatment with napabucasin in vivo [42]. However, their experiments were performed in a microsatellite-instable (MSI) cell line, and tumor growth was only monitored during treatment. In contrast, we used a microsatellite-stable (MSS) cell line, which is characterized by an underlying genetic pathway of the vast majority of sporadic CRC [43]. In addition, we assessed full tumor regrowth to measure treatment response, which more closely mirrors the clinical situation. Consistent with our finding that napabucasin sensitizes esophageal cancer cells to CRT, Ebbing and colleagues recently demonstrated that stroma cell-derived IL-6 mediates CRT resistance of esophageal adenocarcinomas, which could be reverted by inhibition of IL-6 [44]. Based on all these findings, we propose that this knowledge may be translated into a personalized treatment strategy (see Figure 3D) that includes screening of pretherapeutic tumor biopsies for the presence of phosphorylated STAT3, followed by a combined treatment with CRT and napabucasin in case of phospho-STAT3 positivity. Noteworthily, phospho-STAT3 can be detected in up to 40% of CRC [24,45,46] and up to 60% of esophageal cancer [47,48]. Thus, reversing CRT resistance by napabucasin treatment could potentially benefit a large proportion of patients suffering from these cancers.

Mechanistically, phospho-STAT3 executed CRT resistance in rectal cancer cells by supporting NOTCH signaling, specifically by stimulating expression of RBPJ, the key transcriptional effector of the NOTCH pathway. This observation is in accordance with a previous report, showing that JAK-STAT signaling plays a role in the expression of HES1, a prominent NOTCH downstream effector protein [38]. Interestingly, over-expression of HES1 increased STAT3 phosphorylation activity in colon cancer cells [49] and the amount of HES1 also correlated with the activity of STAT3 in our rectal cancer cells, indicating a reciprocal feed-forward regulation of STAT3 activity, NOTCH target genes and their products. We found RBPJ to be a direct target gene of signaling active STAT3, which may explain these observations together with the tumor cell-autonomous processing of NOTCH leading to NICD production and the assembly of the transcriptionally active RBPJ/NICD complex. Further evidence for a tight link between gp130/STAT3 and NOTCH signaling was provided by Yang et al. who demonstrated that Jagged1 regulates expression of STAT3 in platinum-resistant ovarian cancer cells [50]. NOTCH4/STAT3 crosstalk is also important for epithelial-mesenchymal transition of breast cancer cells and NOTCH inhibition reduced the level of activated STAT3 [51].

While inhibition of the NOTCH pathway has already been linked to sensitization of glioblastoma or breast cancer cells to radiation [52], we now show that the NOTCH pathway, controlled by gp130/STAT3 signaling, regulates CRT responsiveness in rectal cancer. This conclusion is supported further by our finding that expression of NOTCH isoforms correlated with the clinical outcome in rectal cancer patients. In accordance with this, NOTCH1 and NOTCH4 were found to be expressed in a subset of breast cancer cells where their expression also correlated with poor prognostic factors [53]. Likewise, in non-small cell lung cancer (NSCLC), expression of NOTCH3 was associated with poor survival of patients [54] while in gastric cancer, high expression of all four NOTCH isoforms correlated with a short relapse-free survival [55].

Collectively, appropriate clinical trials are required to validate the suitability of both our concept to reverse CRT resistance and the value of STAT3 and/or NOTCH as prognostic biomarkers (Figure 3D). However, from a clinical perspective, enhancing responsiveness to CRT will likely increase the fraction of patients that show a pathological complete response after treatment. Potentially, these patients could be spared from the morbidity and mortality of radical surgical resection, which can be substantial, including urinary, sexual or bowel dysfunction in up to 70% of patients, anastomotic leakage in 10–15% of patients, and the necessity for a permanent stoma in up to 20% of the patients [3,5,6]. The clinical concept to omit surgical resection in case of a complete response after CRT, which is referred to as watch-and-wait strategy, is currently intensively and controversially discussed in the field [56,57,58,59]. Nevertheless, it would add a significant clinical as well as socioeconomic benefit for the individual patient and the community, respectively.

## 4. Materials and Methods

### 4.1. Cell Culture, RNA Interference, and Western Blot Analysis

Human rectal cancer cell lines SW837 and SW1463 and the STAT3-negative colon cancer cell line LS411N were obtained from the American Type Culture Collection (Manassas, VA, USA). Human esophageal adenocarcinoma cell lines FLO-1, OAC-P4C, OE-19, OE-33, and SK-GT-4, and squamous cell carcinoma cell lines Kyse-70, Kyse-150, Kyse-180, and Kyse-270 [60] were obtained from the German collection of cell cultures and microorganisms (DSMZ, Braunschweig, Germany). Mycoplasma contamination was routinely tested using MycoAlertVR Mycoplasma Detection Kit (Lonza, Basel, Switzerland), and cross-contamination was surveyed by short tandem repeat (STR) profiling (DSMZ). Transfections with siRNA pools were performed as described [19]. Cell lysis and immunoblotting were performed as described [19]. For detection of phosphorylated STAT3, CRC cells were stimulated either with recombinant IL-6 (Biochrom, Berlin, Germany) or Hyper-IL-6 (Hy-IL-6), while esophageal cancer cells were left unstimulated.

### 4.2. Colony Formation Assay

Adherent cells were incubated over night with 3 µM 5-fluorouracil (5-FU) (Sigma-Aldrich, St. Louis, MO, USA) overnight, followed by X-ray irradiation (0, 1, 2, 4, 6, 8 Gy; Gulmay Medical, Camberley, UK). For pre-treatments napabucasin was administered for 1 h, and Hy-IL-6, tocilizumab or ruxolitinib for 16 h. After cell line-specific incubation times, colonies were stained with Mayer’s hemalum solution (Merck KGaA, Darmstadt, Germany), counted and analyzed [61].

### 4.3. STAT3 Activity and Expression of STAT3 Variants

The activity of STAT3 was measured by dual luciferase reporter (DLR) assays as described before [19]. Site-directed mutagenesis to generate STAT3 mutants harboring amino acid exchanges Y705F, S727A, or Y705F/S727A was performed on wild-type STAT3 cDNA. The integrity of all constructs was confirmed by sequencing. For expression, all STAT3 constructs were ligated in frame into the expression vector pmaxKS, encoding a C-terminal HA peptide tag. Transfection into LS411N cells was done using the Amaxa nucleofection technology (Lonza).

### 4.4. Electrophoretic Mobility Shift Assay (EMSA)

EMSA was performed as described [62]. Briefly, SW837 cells were left untreated or stimulated for 30 min with Hy-IL-6 (20 ng/mL) and were lysed in cytoplasmic extraction buffer. Nuclei were isolated by centrifugation and incubated in 50 μL nuclear extraction buffer. After centrifugation at 16,000× *g* for 15 min and 4 °C, nuclear extracts were mixed with the same amount of cytoplasmic extracts from the same cells. As positive control for GAS binding, lysates of unstimulated or IFN-γ- (50 ng/mL, Biomol, Hamburg, Germany) stimulated HeLa cells were used. Note, HeLa cells coexpress STAT1 and STAT3, which possess distinct electrophoretic mobility, and hence, can be distinguished from each other when simultaneously detected by EMSA. The slower (upper) and faster (lower) migrating band represents STAT3 and STAT1, respectively [36]. For testing of STAT3 binding to the GAS-like element in the *RBPJ* promotor region, we used duplex oligonucleotide probes with 5 bp T overhangs at their 5′ end. Sequences are depicted in Appendix A. End-filling reaction catalyzed by the Klenow fragment (New England Biolabs, Frankfurt am Main, Germany) generated [^33^P]-labeled probes. Four µl of cellular extracts were incubated with 8 µL of EMSA reaction buffer containing 1 ng of the [^33^P]-labeled probes. For competition experiments, a 750-fold molar excess of unlabeled native RBPJ was added to the reaction and incubated for 15 min at room temperature. Following electrophoretic separation on 8% acrylamide:bisacrylamide gels (29:1), DNA-binding complexes were autoradiographically detected on vacuum-dried gels using a Typhoon FLA 9500 laser phosphorimaging system (GE Healthcare, Uppsala, Sweden).

### 4.5. Screening STAT3 Target Genes by RNA-Seq and Opposite Direction Analysis

A detailed description of STAT3 target gene identification is provided in Appendix A. Briefly, STAT3 expression was silenced by siRNA, either without further treatment or after treatment with Hy-IL-6. Total RNA samples were processed for preparation of cDNA libraries exhibiting an average length of 300 bp. Libraries were pooled and sequenced on the Illumina HiSeq 4000 (Illumina, Inc., San Diego, CA, USA). Differentially expressed genes (FDR cut-off: 0.01) were identified for three conditions: siCtrl. vs. siCtrl. + Hy-IL-6, siCtrl. vs. siSTAT3, and siCtrl. + Hy-IL-6 vs. siSTAT3 + Hy-IL-6. Opposite direction analysis (ODA) was employed to identify genes that were significantly upregulated upon Hy-IL-6 stimulation and, inversely, downregulated upon STAT3 silencing. The sequencing data and abundance measurement files have been submitted to the NCBI Gene Expression Omnibus (GEO) under the accession number GSE139455. Semi-quantitative RT-PCR analysis was performed for selected genes as described before [19].

### 4.6. Patients, Gene Expression Profiling and Survival Analysis

This project was conducted within the context of the Clinical Research Unit 179 (KFO179), approved by the Ethics Committee of the University Medical Center Goettingen, and with informed consent obtained from all patients. Detailed information can be found in Appendix A. Briefly, the expression levels of *NOTCH1-4* were extracted from gene expression profiles of pretherapeutic biopsies from 207 patients with locally advanced rectal cancer who were treated with preoperative chemoradiotherapy. For correlation of gene expression data with clinical parameters, Kaplan-Meier curves displaying disease-free survival (DFS) were generated. Patients were grouped according to gene expression levels above or below the median expression of a particular mRNA.

### 4.7. Mice

Animal experiments were approved by the German Animal Welfare Act (reference number: 33.9-42502-04-17/2383). SW1463 cells were subcutaneously injected into the right flank of athymic nude Naval Medical Research Institute (NMRI) Foxn1nu/Foxn1nu mice. Experiments started when the tumor reached a volume of about 150 mm^3^. For CRT/napabucasin treatment, mice were randomly separated into four different groups: DMSO, napabucasin, DMSO + CRT, and napabucasin + CRT. Body weight and tumor volume were measured thrice weekly. According to the legal termination criterion, mice were sacrificed after tumor volume reached approximately 1500 mm^3^.

### 4.8. Statistical Analysis

*p*-values and FDR-values < 0.05 were considered significant. Except for immunoblot analyses, RNA-Seq and EMSA, experiments were performed as technical triplicates and independently repeated at least three times. For data analysis of irradiation experiments, a two-way analysis of variance (ANOVA) was used to calculate significant differences between control and treatment groups and were performed using Microsoft Excel software (version 2016 MSO, Add-in “Data Analysis”, Microsoft Corporation, Redmond, WA, USA). For visualization, data of irradiation experiments are presented as mean and standard error of the mean (s.e.m.) from at least three independent experiments using the software KaleidaGraph (version 4.1.0, Synergy Software, Reading, PA, USA). Statistical analyses of DLR activity were performed using an unpaired two-tailed Student’s t-test in Microsoft Excel and visualized in Grapher software (version 8.2.460, Golden Software, Golden, CO, USA). To calculate *p*-values for correlation of qPCR and RNA-Seq data, Pearson’s correlation was applied in Microsoft Excel. Statistical tests of tumor volume and Kaplan-Meier analysis were performed in GraphPad Prism (version 8), mixed-effects analysis using Tukey’s multiple comparisons test and Log-rank (Mantel-Cox) test, respectively.

## 5. Conclusions

Inflammatory cytokine receptors of the gp130 family synergize with the NOTCH pathway to render tumor cells refractory against chemoradiotherapy. Signaling elements of the two cascades represent candidate biomarkers for patient stratification, which may spare patients with non-responsive tumors from adverse side effects of chemoradiotherapy. Moreover, blocking the tumor cell-intrinsic gp130/NOTCH signal axis may enhance responsiveness to chemoradiotherapy. Collectively, the discovery of a gp130/NOTCH al-liance as a basis of chemoradiotherapy resistance offers a novel treatment concept for pa-tients with rectal cancer, which, if further validated in clinical trials, would add a signifi-cant clinical as well as socioeconomic benefit.

## Figures and Tables

**Figure 1 cancers-13-00455-f001:**
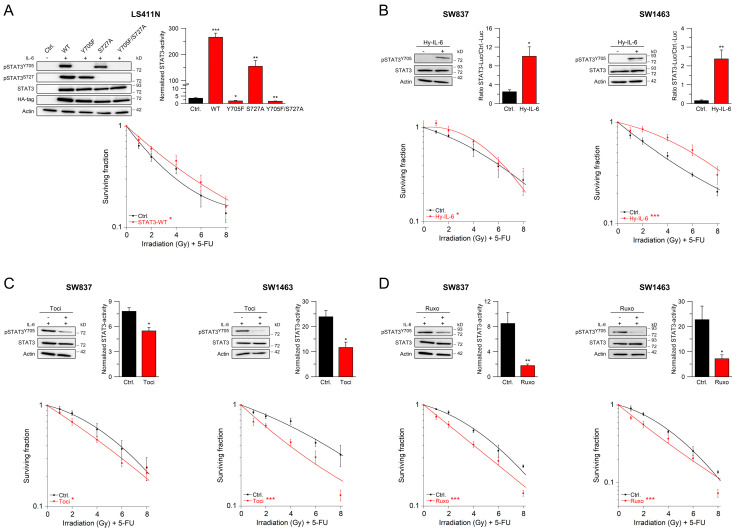
Cytokine receptors of the gp130 family regulate CRT sensitivity via STAT3 activation. (**A**) STAT3-negative LS411N cells were transfected with empty control vector or constructs encoding HA-tagged versions of wild-type STAT3 or STAT3 variants harboring indicated amino acid exchanges. Cells were analyzed for expression and inducible phosphorylation of STAT3 proteins by immunoblotting (upper left) or monitored for inducible STAT3 transcriptional activity (upper right), or were cultured in colony formation assays (CFA) to measure their survival following irradiation in the presence of 5-FU (CRT) (lower graph). (**B**) Hyper-IL-6 (Hy-IL-6)-induced STAT3 phosphorylation and transcriptional activity were analyzed in STAT3-positive SW837 or SW1463 cells (upper panels), and the impact of that stimulation on sensitivity to CRT was assessed (lower panel). (**C**,**D**) cells were treated with (**C**) tocilizumab (Toci) or (**D**) ruxolitinib (Ruxo) and analyzed for STAT3 functionality as described above or were monitored for CFA survival after CRT. Data presented as mean ± s.e.m. from at least *n* = 3 independent biological replicates. * *p* < 0.05, ** *p* < 0.01, *** *p* < 0.001, unpaired two-sample Student’s t-test or two-way analysis of variance (ANOVA). For *p*-values see Appendix A.

**Figure 2 cancers-13-00455-f002:**
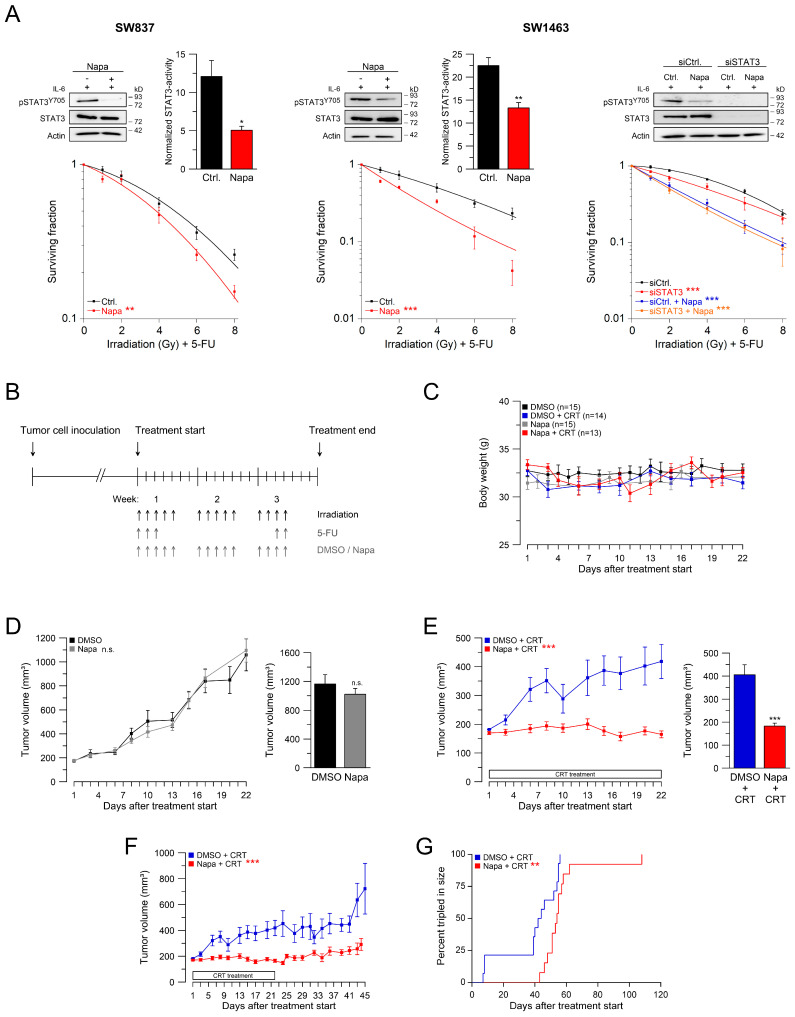
Targeting gp130/STAT3 signaling in vitro and in vivo. (**A**) SW837 or SW1463 cells were left untreated or treated with napabucasin (Napa) (left and middle panel) and analyzed for STAT3 functionality (upper graphs), or were monitored for CFA survival after CRT (lower graphs). Following siRNA-mediated STAT3 silencing and treatment with napabucasin, SW1463 cells or untreated control cells were subjected to STAT3 immunoblot analyses or CFA survival after CRT (right panel). Data presented as mean ± s.e.m. from at least *n* = 3 independent biological replicates. * *p* < 0.05, ** *p* < 0.01, *** *p* < 0.001, unpaired two-sample Student’s t-test or two-way analysis of variance (ANOVA). For *p*-values see Appendix A. (**B**) Scheme of subcutaneous SW1463 rectal cancer xenograft model. (**C**) Body weight curves of mice treated with either DMSO (*n* = 15), napabucasin (*n* = 15), DMSO + CRT (*n* = 14), or napabucasin + CRT (*n* = 13). (**D**,**E**) Tumor volumes of mice during treatment (left panels), and at the end of treatment (right panels). Tumors were treated with DMSO or napabucasin, either without CRT (**D**) or with CRT (**E**). *** *p* < 0.0001 (**E**, left panel), *** *p* = 6.668 × 10^−5^ (**E**, right panel). (**F**) Tumor volumes of mice until day 45 after start of treatment, *** *p* < 0.0001. (**G**) Kaplan-Meier estimates of the time to tumor tripling. Median times to tumor tripling were 43 days (DMSO + CRT) and 54 days (napabucasin + CRT), ** *p* = 1.13 × 10^−2^. *p*-values were calculated by mixed-effects analysis using Tukey’s multiple comparisons test (**D**,**E**, left and **F**), unpaired two-sample Student’s *t*-test (**D**,**E**, right), or Log-rank (Mantel-Cox) test (**G**), see Appendix A. Data points consisted of at least seven mice.

**Figure 3 cancers-13-00455-f003:**
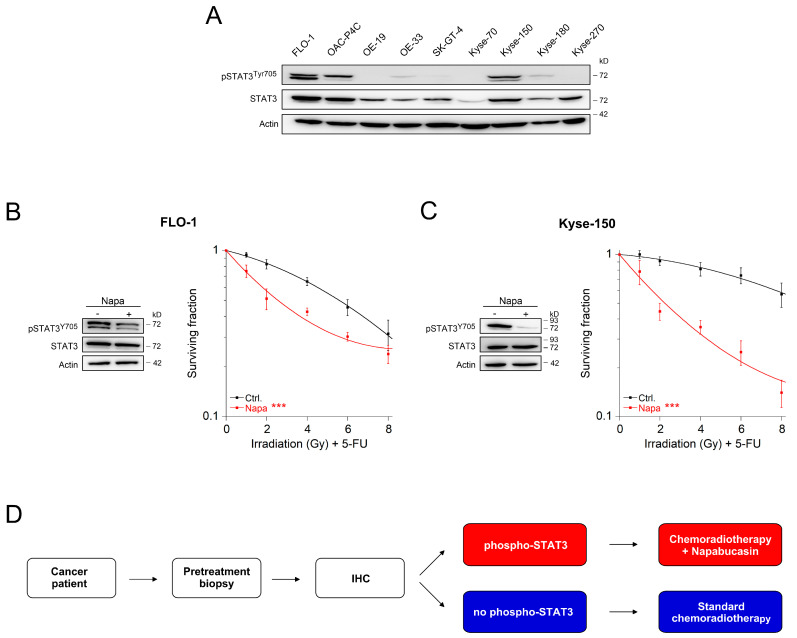
Targeting gp130/STAT3 signaling in esophageal cancer cells. (**A**) Nine esophageal cancer cell lines were analyzed for expression and phosphorylation of STAT3 by immunoblotting. (**B**,**C**) FLO-1 or Kyse-150 cells were left untreated or treated with napabucasin (Napa). Cells were either subjected to Western blot analysis (left panel), or were monitored for CFA survival after CRT (right panel). Data presented as mean ± s.e.m. from at least *n* = 3 independent biological replicates. *** *p* < 0.001, two-way analysis of variance (ANOVA). For *p*-values see Appendix A. (**D**) Putative personalized treatment strategy for cancer patients who are referred to preoperative or definitive CRT. Pretherapeutic biopsies are tested for phospho-STAT3 via immunohistochemistry (IHC). Patients with pSTAT3-positive tumors are treated with a combination of CRT and napabucasin, while pSTAT3-negative tumors are treated with standard CRT.

**Figure 4 cancers-13-00455-f004:**
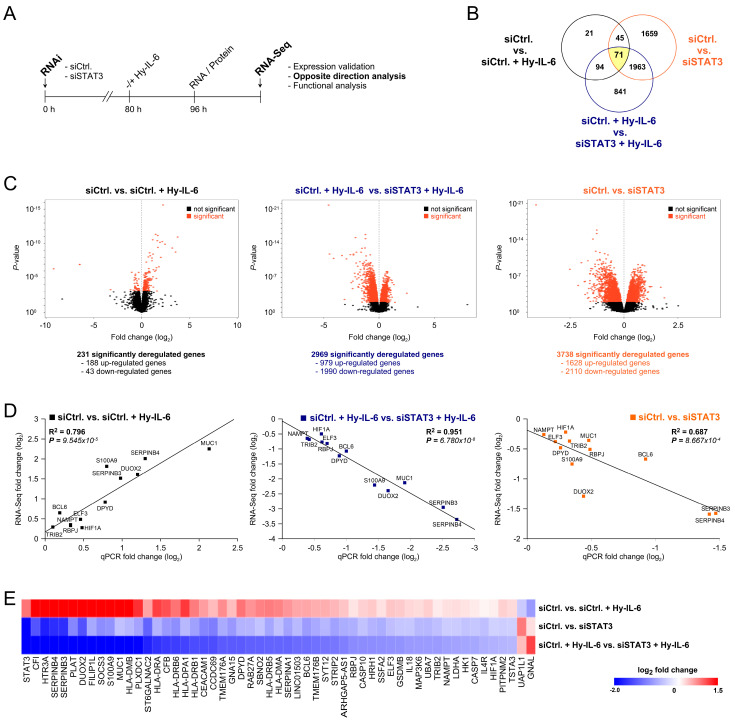
Target genes of the gp130/STAT3 axis. (**A**) RNA-Seq-based detection of STAT3 target genes in SW837 cells with basic or silenced STAT3 expression in the presence or absence of Hyper-IL-6. (**B**) Venn diagram of differentially expressed genes under indicated conditions (*n* = 3). RNA-Seq revealed 231 (siCtrl. vs. siCtrl. + Hy-IL-6, left panel), 2969 (siCtrl. + Hy-IL-6 vs. siSTAT3 + Hy-IL-6, middle panel), and 3738 (siCtrl. vs. siSTAT3, right panel) significant genes (FDR < 0.05), respectively. (**C**) Volcano plots depicting the number and distribution of differentially up- and down-regulated genes (FDR < 0.05, red dots). (**D**) Linear model analysis correlating mRNA fold changes elucidated by RNA-Seq with qPCR values. *p*-values were calculated using Pearson’s correlation. (**E**) Expression profiles of genes fulfilling the Opposite Direction Analysis criteria of being upregulated on stimulation with Hyper-IL-6, and downregulated on STAT3 inhibition, and vice versa (for further information, see Appendix A).

**Figure 5 cancers-13-00455-f005:**
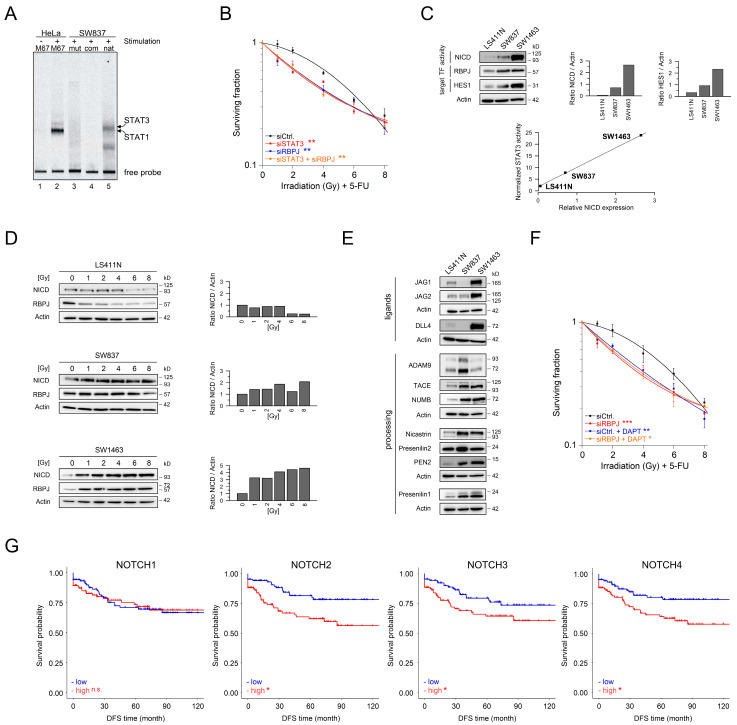
The gp130/STAT3 axis connects with the RBPJ/NOTCH pathway. (**A**) Binding of STAT proteins to [^33^P]-labeled oligonucleotides encompassing prototypic GAS (M67), a mutated GAS (mut) or the native GAS element (nat) from the *RBPJ* promotor, was analyzed by EMSA using extracts of unstimulated or IFN-γ-stimulated HeLa cells, or Hyper-IL-6-stimulated SW837 cells. The radioactively labeled M67 probe was outcompeted (com) by incubating lysates with an excess of unlabeled M67 probe. Labeling of STAT1 and 3 is based on their documented migration pattern in double-positive cells in which STAT3 can be distinguished from STAT1 by its slower electrophoretic mobility [36] (**B**) CFA of SW837 cells after RNAi against *STAT3* and *RBPJ*, either alone or in combination. (**C**) Expression analysis of NOTCH pathway components. (**D**) Immunoblotting of NICD and RBPJ presence following irradiation. (**E**) Expression analysis of γ-secretase complex and additional NOTCH pathway components. (**F**) CFA of SW837 cells after RNAi against *RBPJ* and after treatment with the γ-secretases inhibitor DAPT, either alone or in combination. (**G**) Survival curves of 207 rectal cancer patients who were treated with preoperative CRT. Survival data were plotted against pretherapeutic gene expression levels of *NOTCH1-4*, respectively. Data of CFA experiments presented as mean ± s.e.m. from at least *n* = 3 independent biological replicates. * *p* < 0.05, ** *p* < 0.01, *** *p* < 0.001, two-way analysis of variance (ANOVA). For *p*-values see Appendix A.

## Data Availability

RNA-Seq data are available at the Gene Expression Omnibus (GEO) repository under the accession number GSE139455.

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
