# Peer review of "NOTCH Activation via gp130/STAT3 Signaling Confers Resistance to Chemoradiotherapy"

_cancers, 2021, doi:10.3390/cancers13030455_

Round 1

Reviewer 1 Report

NOTCH activation via gp130/STAT3 signaling confers resistance to chemoradiotherapy. Koerdel et al. manuscript is well written and the overall findings are very sound. Have minor English editing. All the figures need to make clear and visible seems some figures are blurry. 

Author Response

We thank reviewer 1 for taking her/his time to critically evaluate our manuscript and for the valuable recommendation to improve some of our figures.

Specific point:

All the figures need to make clear and visible seems some figures are blurry.

All figures are now provided as high-resolution graphics.

Reviewer 2 Report

In this manuscript, Koerdel et al explore the mechanism of chemoradiotherapy resistance conferred by STAT3 signaling, an hypothesis generated by some of their previous work.  The authors convincingly demonstrate that STAT3 signaling directly upregulates NOTCH signaling by activating the transcription of RBPJ. The authors further demonstrate that regulation of RBPJ independent of STAT3 phenocopies manipulation of STAT3 signaling itself. They further show that this effect is not limited to colorectal cancer, but is also observed in esophageal cancer cells. The possible translational impact of this study is also made clear.

Overall, this manuscript was a joy to review, as it followed a very logical progression of experiments, with careful attention to appropriate controls. The analysis of RBPJ expression with both up and down regulation of STAT3 signaling was especially appreciated.

Minor comment:

 The labels in the EMSA in Figure 5A indicating STAT1 and STAT3- by what criteria is this distinction made? Ideally the identity of STAT3 would be determined by antibody supershift. But between the consensus sequence, appearance of binding after specific pathway stimulation, and the RNAseq data, I do not consider this experiment a requirement, as long as the authors can explain the labelling. 

Author Response

We thank reviewer 2 for taking her/his time to critically evaluate our manuscript and for the enthusiastic comments, which are highly appreciated.

Specific point:

The labels in the EMSA in Figure 5A indicating STAT1 and STAT3 - by what criteria is this distinction made? Ideally, the identity of STAT3 would be determined by antibody supershift. But between the consensus sequence, appearance of binding after specific pathway stimulation, and the RNAseq data, I do not consider this experiment a requirement, as long as the authors can explain the labelling.

We agree that identification of STAT3 by EMSA antibody supershift would further confirm our consensus sequence analysis, the mutational analysis of the STAT3 pathway and our RNAseq data. We also agree that the labeling in Figure 5 needs additional explanation and thank reviewer 2 for his/her recommendation.

The labeling in Figure 5A is based on the documented migration pattern of STAT1 and STAT3 in cells that express both proteins (Sasse et al., new reference #36). In such double-positive cells like HeLa, STAT1 can be distinguished from STAT3 by its slightly higher electrophoretic mobility. Hence, when both proteins are simultaneously detected, e.g. by radioactively labeled GAS probes in EMSA, the slower (upper) and faster (lower) migrating band represents STAT3 and STAT1, respectively. We have included that explanation and the corresponding reference into the revised version of our manuscript.